# BATCH NORMALIZATION AND BOUNDED ACTIVATION FUNCTIONS

## ABSTRACT

Since Batch Normalization was proposed, it has been commonly located in front of activation functions, as proposed by the original paper. Swapping the order, i.e., using Batch Normalization after activation functions, has also been attempted, but it is generally not much different from the conventional order when ReLU is used. However, in the case of bounded activation functions like Tanh, we discovered that the swapped order achieves considerably better performance on various benchmarks and architectures than the conventional order. We report this remarkable phenomenon and closely examine what contributes to this performance improvement in this paper. One noteworthy thing about swapped models is the extreme saturation of activation values, which is usually considered harmful. Looking at the output distribution of individual activation functions, we found that many of them are highly asymmetrically saturated. The experiments inducing a different degree of asymmetric saturation support the hypothesis that *asymmetric saturation helps improve performance*. In addition, we found that Batch Normalization after bounded activation functions has another important effect: it relocates the asymmetrically saturated output of activation functions near zero. This enables the swapped model to have higher sparsity, further improving performance. Extensive experiments with Tanh, LeCun Tanh, and Softsign show that the swapped models achieve improved performance with a high degree of asymmetric saturation.

## 1 INTRODUCTION

Batch Normalization (BN) has become a widely used technique in deep learning. It was proposed to address the internal covariate shift problem by maintaining a stable output distribution among layers. The characteristics of the output distribution of weighted summation operation, which is a symmetric, non-sparse, and "more Gaussian" (Hyvärinen & Oja, 2000), Ioffe & Szegedy (2015) placed the BN between the weight and activation function. Thus, the "weight-BN-activation" order, which we call "Convention" in this paper, has been widely used to construct one block in many architectures (Simonyan & Zisserman, 2014; Howard et al., 2017). "Swap" models, swapping the order of BN and the activation function in a block, have been also attempted but no significant and consistent difference between the two orders has been observed in the case of ReLU. For instance, Hasani & Khotanlou (2019) evaluated the effect of position of BN in terms of training speed and concluded that there is no clear winner and the result depends on the datasets and architecture types.

However, in the case of bounded activation functions, we empirically found that Swap order exhibits substantial improvements in test accuracy than the Convention order with diverse architectures and datasets. We investigate the reason for this accuracy difference between the Convention and the Swap model with bounded activation function based on empirical analysis. For simplicity, our analyses are mainly conducted on Tanh model, but applicable to similar antisymmetric and bounded activation functions. We presents the results with LeCun Tanh and Softsign at the end of the experimental section.

One key difference between Swap and Convention models is the distribution of activation values, as shown in Figure 1. In the Swap model, most activation values are near the asymptotic values of the bounded activation function, that is, highly saturated. This is unanticipated since it is a common belief that high saturation should be avoided. To investigate this paradox, we took one step further

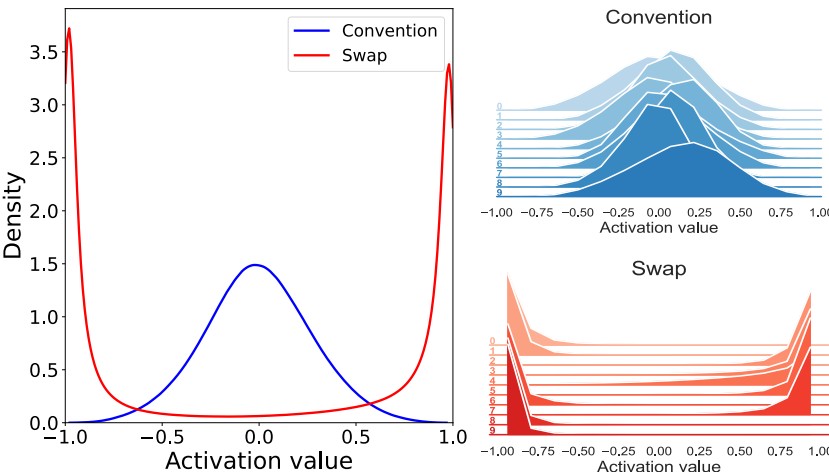

Figure 1: The activation distributions of a layer are almost symmetric (left) in both Convention and Swap models with Tanh. However, the activation distributions of channels in the layer are quite different. Symmetric distributions similar to that of the layer appeared similar to layer distribution in channels in the Convention model (right top). On the other hand, the Swap model have a one-sided distribution of boundary (bottom right). We chose ten consecutive channels from the 8th layer of the VGG16 model trained on CIFAR-100.

and looked at the output distribution of individual activation functions, not just a whole layer. To our very surprise, even though the distribution is fairly symmetric at the layer level, the activation values of each channel are biased toward either one of the asymptotic values, or *asymmetrically saturated*. We assume that this asymmetric saturation is a key factor for the performance improvement of the Swap model since it enables Tanh to behave like a one-sided activation function. In the experiments we designed to examine whether asymmetric saturation is related to the performance of models with bounded activation functions, we can observe that the accuracy and the degree of asymmetric saturation are highly correlated.

BN after Tanh does not just incur asymmetric saturation but also shifts the biased distribution near zero, which has the important effect of increasing sparsity. Sparsity is generally considered to be a desirable property. For instance, Glorot et al. (2011) studied the benefits of ReLU compared to Tanh in terms of sparsity. One thing to note is that if each channel is symmetrically saturated, BN will not increase sparsity much since the mean is already close to 0. In contrast, the one-sided property of asymmetric saturation causes at least half of the sample values after normalization to be almost zero, allowing the Swap model to have even higher sparsity than the Convention model. Ramachandran et al. (2017) explored novel activation functions by an automatic search for different activation functions. The top activation functions found by search are one-sided, and the boundary value is near zero, similar to ReLU. The penalized Tanh activation (Xu et al., 2016), inserting leaky ReLU before Tanh, also introduces skewed distribution, and the penalized Tanh achieved the same level of generalization as ReLU-activated CNN. Analogous to the activation functions found in the previous studies, asymmetric saturation combined with normalization makes a bounded activation function behave much like ReLU, achieving comparable performance.

Our findings are as follows:

- The Swap model using Batch Normalization after bounded activation functions performs better than the Convention model in many architectures and datasets.

- We discover the *asymmetric saturation* at the channel level and investigate its importance through carefully-designed experiments.

- We identify the high sparsity induced by Batch Normalization after bounded activation functions and perform an experiment to examine the impact of sparsity on performance.

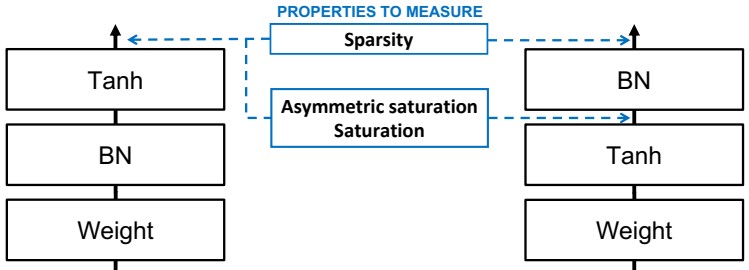

Figure 2: Illustration of Block designs of the Convention order (left) and Swap order (right), and locations for property measurement.

## 2 SETTINGS FOR INVESTIGATION AND NOTATION

**Models.** The main purpose of the investigation is to analyze the benefits of using BN after bounded activation functions, more specifically, a bounded activation function that is an odd function and has two boundaries. We examine the VGG-like model trained on CIFAR-100 with replacing the activation function from ReLU to Tanh. However, because the VGG architecture was proposed for the ImageNet dataset, the model is overparameterized for the CIFAR dataset. It incurs poor performance and difficulty to investigate the Swap model. Thus, we cut out the last convolution layers and select the best model based on the validation accuracy. The model with five cut-out layers shows the best accuracy as in Appendix A.6. We call this model "VGG16_11" and use this architecture to investigate Conv and Swap orders. Although a VGG11 model has already been proposed in Simonyan & Zisserman (2014), the validation accuracy of VGG16_11 is significantly higher than VGG11 (VGG11: Conv 64.55%, Swap 69.94, VGG16_11: Conv 69.5%, Swap 74.11% ). At inference time, The BN normalizes the input distribution to have zero-mean and unit-variance by using the running statistics (e.g., $\hat{\mu}$ for running mean and $\hat{\sigma}$ for related to running variance), and then applies the affine transformation, which has a scaling parameter $\gamma$ and a shifting parameter $\beta$. The Convention model normalizes the outputs of the weighted summation operation conducted in the weight layer, and then Tanh activates the block outputs. On the other hand, in the Swap model, Tanh directly activates the weight layer outputs, and then BN is applied to generate block outputs.

**Metrics.** We consider 3 properties to investigate each order: saturation, asymmetric saturation, and sparsity. We measure the degree of saturation at the outputs of Tanh in the layer units. To measure the asymmetric saturation, we collect the outputs of Tanh in channel units. For the sparsity measure, we collect the outputs of each block in the channel units. Layer structure and measurement locations are illustrated in Figure 2.

**Setups for experiment.** For the experiment in Section 4.2, the weight decay on the convolution layer is fixed, and we vary the weight decay intensity on BN. This experiment's learning rate and the convolution layer's weight decay followed the NWDBN model's hyperparameters. NWDBN is the Convention based model, but the affine parameters of BN are zero. Based on these hyperparameters, we increase the intensity of weight decay on $\beta$ in BN from 0.0 to 0.001 by 0.0001. For the experiment in Section 5.3, the learning rate and convolution layer's weight decay followed the Swap model's hyperparameters. Then, we change the weight decay intensity on the affine transformation parameters in BN. The intensity list of weight decay are 0, 1e-6, 5e-6, 1e-5, 5e-5, 1e-4, and 5e-4. For the experiment in Section 7.1, we train models on 4 benchmarks (CIFAR-10, CIFAR-100, Tiny ImageNet, and ImageNet), 2 base-architectures (VGG16_11, MobileNet), and 2 activation functions (ReLU, Tanh). Because Tanh has non-linearity in everyplace except the origin, it can not follow the design of residual connection proposed in He et al. (2016). Thus, we choose architectures where a skip connection does not exist. For the experiment in Section 7.2, we trained VGG16_11 with 3 activation functions (Tanh, Lucun Tanh, Softsign) on CIFAR-100 dataset. All results except the ImageNet dataset are conducted on 3 random seeds and averaged over seeds for all the measure values and accuracy. We use the SGD optimizer, weight decay regularization, and a 2-step learning rate decaying strategy that decays by 0.1. We conduct a grid search to obtain the best model for investigation. We explore learning rate and weight decay. The hyperparameters that we use are demonstrated in Appendix A.1.

## 3 OVERLY SATURATED TANH BUT WELL-GENERALIZED MODEL

Saturation refers to a situation where most of the outputs of bounded activation functions are close to the asymptotic value of the function. When training a neural network with a bounded activation function whose center is the origin, the output increases due to the weight gradually increasing. The increased output values map close to the near asymptote in bounded activation functions, as shown in the experiment in Glorot & Bengio (2010). Thus, saturation is bound to occur. However, excessive saturation results in a gradient vanishing problem. The gradient of points near the asymptotic values is almost 0. Therefore, the gradients of saturated activations vanished. Various methods were proposed to prevent excessive saturation. Glorot & Bengio (2010) proposed an initialization scheme, Rakitianskaia & Engelbrecht (2015a;b) proposed a metric to measure the degree of saturation for monitoring the training, Bhat et al. (1990) pre-scaled the inputs of the activation function, and Chen & Chang (1996) proposed adaptable bounded activation.

### 3.1 SATURATION METRIC

We introduce a saturation metric based on how closely outputs the values to the maximum absolute value of the output range of the function. The target outputs for measuring the saturation $G^l = [g_1^l, g_2^l...g_N^l] \in \mathbb{R}^N$ is the flattened outputs of $l^{th}$ layer in fully-connected block or convolution block. $N$ is $SD^l$ for fully-connected blocks and $SC^lH^lW^l$ for convolution blocks, where $S$ denotes the total number of test samples, $D^l$ denotes the dimension size of layer outputs in $l^{th}$ fully-connected block, and $C^l$, $H^l$, $W^l$ respectively denotes the number of channels, height, and width in $l^{th}$ convolution block. We take the absolute value of the input and divide it by the maximum absolute value to normalize it to [0, 1]. The formulation for normalization of $i^{th}$ element in $l^{th}$ layer feature map, $\hat{g}_i^l$, is as follows:

$$\hat{g}_i^l = \frac{|g_i^l|}{\tilde{g}^l}, \tag{1}$$

where $\tilde{g}^l \in \mathbb{R}$ is the maximum absolute value of $G^l$. Since the possible output range of the bounded function is fixed. We use the absolute asymptotic value of the bounded function as a all element of $\tilde{g}^l$ for measuring saturation. For instance, we set $g_d^l$ to 1 for the Tanh model. We averaged all the normalized values in a layer for our saturation metric. The formulation of our saturation metric on $l^{th}$ layer, $t^l$, is as follows:

$$t^l = \frac{\Sigma_{i=1}^N \hat{g}_i^l}{N}. \tag{2}$$

$t^l$ has the range of $[0, 1]$; it approaches 1 if $G^l$ is highly saturated as illustrated in Appendix A.3. Also, as an implementation issue, the calculation was performed in units of mini-batch, and the details are described in appendix A.11.

### 3.2 HIGH SATURATION IN THE SWAP MODEL

Even if only the layer order was changed from the Convention order to the Swap order, there was a 4.61%p test accuracy improvement. The results of this model and other models can be found in Table 1. However, when we measure the layer saturation in both models, the Swap model has highly saturated layers. The maximum saturation of the Swap model (**0.86**) is significantly higher than the Convention model (**0.45**). The saturation of the Swap model shows over 0.7 in almost half of the layers. Even more, some layers are overly saturated at almost 0.86. On the other hand, the saturation of the Convention model is lower than 0.5 over all layers. (Figure 3) This is counter-intuitive as excessive saturation is considered an undesirable situation in the previous works.

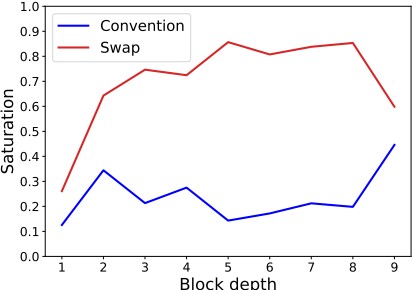

Figure 3: Layer Saturation of Convention and Swap models

## 4 ASYMMETRIC SATURATION

Our saturation metric can dismiss the channel properties due to the summarization of channels in the layer. Thus, we conduct channel inspection. Interestingly, when we examine channel distribution, the saturation in that layer has biased to one asymptotic value. Asymmetric saturation appears in most channels on the excessively saturated layer in the Swap model. In contrast, the channel distribution of the Convention is almost zero centralized.

### 4.1 ASYMMETRIC SATURATION METRIC

The target outputs for measuring the asymmetry $Q^{l,c} = [q_1^{l,c}, q_2^{l,c}...q_M^{l,c}] \in \mathbb{R}^M$ is the flattened activation outputs of $l^{th}$ layer and $c^{th}$ dimension for fully-connected block or $c^{th}$ channel for convolution block. $M$ is $S$ for fully-connected blocks and $SH^lW^l$ for convolution blocks. To measure the channel asymmetry more precisely, we introduce skewness, the metric for measuring the asymmetry. The formulation of the sample skewness for $l^{th}$ layer and $c^{th}$ channel , $k^{l,c}$, is as follows:

$$k^{l,c} = \frac{\sqrt{M(M-1)}}{M-2} \frac{\frac{1}{M}\Sigma_{i=1}^M(q_i^{l,c}-\mu^c)^3}{[\frac{1}{M}\Sigma_{i=1}^M(q_i^{l,c}-\mu^c)^2]^{\frac{3}{2}}}, \tag{3}$$

where $\mu^c \in \mathbb{R}$ is the mean of $l^{th}$ layer and $c^{th}$ channel's activation outputs. The skewness value has directional distribution information, negative for left-skewed and positive for right-skewed. However, we want to measure asymmetry regardless of direction. Thus we take the absolute value to remove the directional information. The metric for the layer skewness, $k^l$, is as below:

$$k^l = \frac{1}{C}\Sigma_{i=1}^C|k^{l,i}|. \tag{4}$$

The layer distributions in both Convention and Swap models are symmetry, but the channel distributions are quite different. Thus, we measure the asymmetry on channel-wise, not layer-wise, like the saturation metric. As an implementation issue, the calculation was performed in units of mini-batch, and the details are described in appendix A.11.

As shown in Figure 4, All of the layer skewness in the Convention model measured close to 0. Therefore there has little asymmetric distribution. However, in the Swap model, the skewness of layers is relatively higher than in the Convention model. Furthermore, the skewness values are high along the high saturation blocks. It, therefore, implies that saturation occurs with asymmetry. The relationship between our skewness metric and the different distribution shapes is illustrated in Appendix A.3.

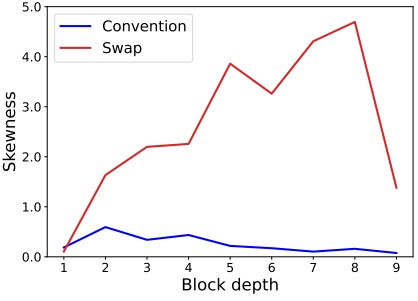

Figure 4: Layer Skewness in Convention and Swap models

### 4.2 EFFECT OF ASYMMETRIC SATURATION ON GENERALIZATION PERFORMANCE

In order to demonstrate the effectiveness of asymmetric saturation, we introduce a method to control the level of asymmetry in the Convention model. First, let us organize the reason why the Convention model cannot make use of asymmetric saturation. We assume that the Convention model can not generate asymmetric saturation well due to the weight decay effect on affine transform parameters in BN. In the experiment to verify the mean and variance effects on skewness, we can confirm that both statistical values, the mean and variance of Tanh input, affect asymmetry on Tanh output. The skewness value of Tanh's output on the different input mean and standard deviation can be found in Appendix A.4. From this perspective, the affine parameters with weight decay generate the input of Tanh to utilize the center of Tanh by decreasing the mean and variance of its input. Thus, it could decrease the asymmetry of the Tanh output. Therefore, we train a model with no weight decay on BN to encourage asymmetric saturation in the Convention model. As a result, the NWDBN model shows improved accuracy of **72.27%** compared to the Convention model **69.5%**. To closely examine the effects of asymmetric saturation on test accuracy,

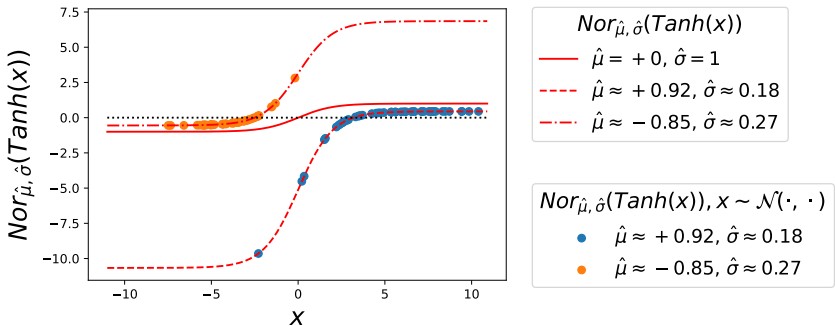

Figure 6: Shapes of combined Tanh with normalization functions, and samples related to BN statistics. The functions are plotted as lines, and the samples are plotted as dots. We choose some normal distributions whose samples generate $\hat{\mu}$ and $\hat{\sigma}$ after the Tanh and randomly generate input samples for Tanh. Note that the $\hat{\mu}$ and $\hat{\sigma}$ are the statistics of Tanh output in the Swap order.

we increase the intensity of weight decay on the Beta parameter, which can eliminate the biasing of the asymmetric saturation in the NWDBN model. As shown in Figure 5, increasing weight decay intensity decreases the skewness in the NWDBN model. Additionally, the test accuracy decreased along with the skewness.

# 5 SPARSITY

## 5.1 ASYMMETRIC SATURATION WITH BATCH NORMALIZATION CAN INDUCE HIGH SPARSITY

Sparsity is a desirable property in deep learning. One of the successes of the method that introduces a sparsity is the Relu. ReLU achieves a high generalization performance by utilizing the strengths of sparsity (Glorot et al., 2011). The sparsity of ReLU is due to the one boundary placed at 0. Thus ReLU activates all negative inputs to 0. The other work that shows the advantage of having one asymptote at 0 is Ramachandran et al. (2017). They conducted an automatic search strategy to look up various activation functions used. The top prominent activation functions identified through search are one-sided with a boundary value close to zero, like ReLU. Also, Xu et al. (2016) introduced penalized Tanh activation, which places leaky ReLU before Tanh to enhance the performance of Tanh, which perform as well as ReLU and introduce asymmetry in Tanh.

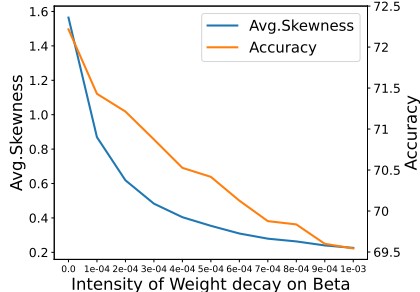

Figure 5: Relation between accuracy and averaged skewness over layers. The "Avg.Skewness" averaged all the layer-wise skewness in each model with different weight decay intensity. The NWDBN model is denoted as 0.0 intensity in the graph.

We found that the Swap model also can increase the sparsity by shifting the majority of the values to 0 when asymmetric saturation occurs. The normalization in BN makes the distribution to be zero mean. When the asymmetric saturation occured on precede Tanh, the majority of activations are saturated on one side of Tanh output. Thus, the normalization applied on this distribution the majority of values are shifted to near zero which incurs a increasement of sparsity.

## 5.2 SPARSITY COMPARISON

The NWDBN model shows better performance than the Convention model by inspiring the asymmetry, but it underperforms the Swap model. We found that the rise of asymmetric saturation in the NWDBN model gives a benefit in terms of asymmetry but decreases the sparsity. In other words,

increased asymmetry of activations in the Convention model generates more activation values close to -1 or 1, which incurs less sparse block output. Based on this intuition, we hypothesize that the Swap model has strength on sparsity. To compare the models, we introduce our sparsity metric to verify the sparsity on each model.

We leverage our saturation metric and modify it for the sparsity metric. Our saturation metric measures the degree to which many values are saturated with the maximum value. On the other hand, sparsity is measured by how a small number of coefficients contain a large proportion of the energy. The more saturated the distribution, the more coefficients divide the total energy. In short, higher saturation decreases sparsity. Therefore, the sparsity metric can be regarded as the reverse of the saturation metric. However, there is differences to the saturation metric. Whereas the saturation is measured on the output of Tanh, sparsity is measured on the output of the blocks, i.e., the sparsity of the Conv model is measured on the Tanh output, and the sparsity of the Swap model is measured on the BN output. Thus, for the measuring the sparsity, we modify $\tilde{g}^l$ in Equation 1 to the vector of maximum absolute output in unit-wise, $\bar{\mathbf{g}}^l \in \mathbb{R}^{D^l}$ for fully connected block and $\mathbb{R}^{C^l}$ for convolution block. Then, we normalize $|g_i^l|$ by the corresponding unit value in $\bar{\mathbf{g}}^l$. The formulation of modified normalized element , $\dot{g}_i^l$, is $|g_i^l|/\bar{g}_d^l$, where $d$ is the corresponding dimension or channel index of $i^{th}$ output. Consequently, the modified saturation metric, $\bar{t}^l$, is $\Sigma_{i=1}^N \dot{g}_i^l/N$ and our sparsity metric for $l^{th}$ layer, $s^l$, is $1 - \bar{t}^l$. Also, we investigated how our sparsity metric satisfies the conditions of the sparsity metric. We demonstrate our sparsity metric based on the 6 desired heuristic criteria of sparsity measures described in Hurley & Rickard (2009). Our sparsity metric satisfies 5 criteria among 6 criteria. The proof can be found in Appendix A.10.

We first measured saturation on each model's block output to measure the sparsity and subtracted the saturation value from 1. Then, averaged the sparsity over layers. The sparsity of each model is as follow: Convention (**0.717951**), NWDBN (**0.287974**), Swap (**0.848927**). The Swap model shows the largest sparsity. The result also shows that the Convention model can generate sparse distribution. Because of the weight decay on BN, a zero-centered distribution insert to the Tanh in Convention model. Lastly, as we expect, the NWDBN model shows the lowest sparsity. However, Since the NWDBN model has a higher asymmetry than the Convention model, the NWDBN model can outperform the Convention model.

### 5.3 Effect of Sparsity on Generalization Performance

In this section, we encourage the sparsity in the Swap model and investigate its effects on test accuracy. As mentioned in Section 5.1, the Swap order can enhance the sparsity when asymmetric saturation occurs. This sparsity can be promoted in training by affine parameters in BN. Decaying on affine parameters gathers the most values to 0 during the training phase. Note that the normalization operation shifts the majority near zero, and affine transformation imposes the majority of distribution more centered to 0. To enhance the sparsity of the Swap model, we increase the weight decay of affine transformation parameters. The larger weight decay may further increase the sparsity of BN output. As shown in Figure 7, the increase in the model's sparsity and accuracy are highly correlated.

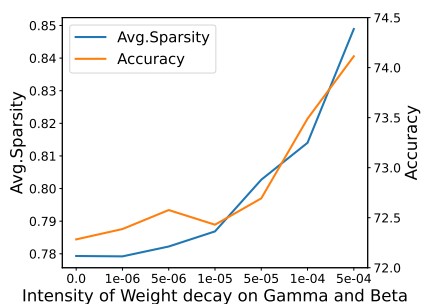

Figure 7: Influence of sparsity on accuracy. we measure the averaged saturation over layers in the Swap model trained with each random seed and calculate the sparsity by our sparsity metric.

## 6 Summary of the Main Analysis

We trained 3 types (Convention, NWDBN, Swap) of models in the above analysis experiments. Each model creates a different output distribution of layers due to differences in structure and regularization effects. Output distributions of these models are described in Figure 8. The Convention model, which is illustrated in Figure 8 (top), normalizes extracted features from the convolution

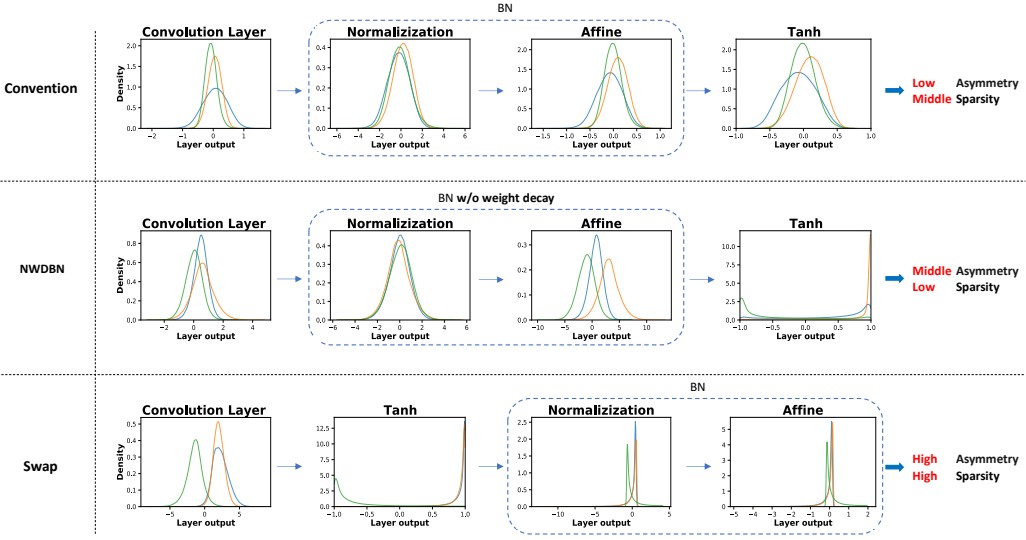

Figure 8: The distribution of VGG16's 5th block's output on randomly chosen 3 channels. We chose a block where all 3 models were considerably saturated. All test samples in the CIFAR-100 dataset are used to construct the distribution.

Table 1: Test accuracy with different activation functions and layer orders for VGG16 and MobileNet.

| Dataset | VGG16 Tanh | | MobileNet Tanh | | VGG16 Relu | | MobileNet Relu | |
|---|---|---|---|---|---|---|---|---|
| | Convention | Swap | Convention | Swap | Convention | Swap | Convention | Swap |
| CIFAR-10 | 91.75 | 92.90 | 91.54 | 92.48 | 93.69 | 93.04 | 92.2 | 91.93 |
| CIFAR-100 | 64.84 | 72.17 | 64.47 | 70.63 | 73.68 | 71.79 | 70.06 | 69.49 |
| Tiny ImageNet | 49.29 | 57.05 | 50.85 | 51.79 | 61.54 | 59.045 | 59.79 | 59.1 |
| ImageNet | 60.85 | 67.04 | 64.26 | 72.07 | 73.83 | 72.95 | 70.48 | 71.1 |

layer. After that, affine parameters are applied to the normalized features. These affine parameters generate zero centralized activation caused by the effect of weight decay. The NWDBN order also normalizes the extracted feature from convolution layer. Still, Unlike the Convention model, there are no downscaling effects on affine transform parameters. For this reason, the input distribution to Tanh can generate a distribution away from zero and produce a relatively high asymmetry distribution than the Convention model. We can observe that asymmetric saturation is generated through Tanh in Figure 8 (middle). However, the asymmetric saturation in the NWDBN model leads to low sparsity, which negates the benefits of sparsity. Far from the above models, the Swap model applied Tanh to the extracted features from convolution layer, and BatchNorm is followed. Therefore, if Tanh generates asymmetric saturation, then it could be a significant number of activations will be moved near zero, helping to increase sparsity. The layer output distribution can be found in Figure 8 (bottom).

# 7 EXTENDED EXPERIMENTS

## 7.1 RESULTS ON VARIOUS DATASETS AND ARCHITECTURES

We mainly investigated VGG16_11 with Tanh model trained on CIFAR-100 dataset. In this section, we adopt Swap order on varied settings, which are various datasets (CIFAR-10, CIFAR-100, Tiny ImageNet, ImageNet), architectures (VGG, MobileNet), and activation functions (ReLU, Tanh).

The Swap order and the Convention order of the ReLU model do not show a large difference in generalization performance than the difference of Tanh model, and this could be ReLU has the

Table 2: VGG16_11 with bounded activation functions on CIFAR-100, we used averaged skewness over layers for calculating the difference of skewness.

| Activation | Order | | Swap - Convention |
|---|---|---|---|
| | Convention | Swap | $\Delta$Avg.Skewness |
| Tanh | 69.5 | 74.11 | 2.38 |
| LeCun Tanh | 67.82 | 74.46 | 1.90 |
| Softsign | 70.01 | 73.65 | 1.28 |

structural ability to produce asymmetric and sparse activations. However, in the case of Tanh, every model with Swap order outperforms the Convention ordered models with significant generalization improvement. The Convention order slightly performs better than the Swap order except for the ImageNet dataset on ReLU model. The Swap MobileNet with Tanh especially performs better than the Convention Mobilenet with ReLU on CIFAR and ImageNet datasets. The results can be found in Table 1. Also, all Swap models generate asymmetry on Tanh.

The asymmetric saturation tends to occur from the front layers. Also, we can find that the range of the asymmetric saturation existence block is related to the amount of dataset information and dataset resolution. For example, when comparing the CIFAR-10 and CIFAR-100, the asymmetrically saturated layers happen further back. When comparing the Tiny ImageNet, and ImageNet, the model trained on the ImageNet generates asymmetric saturation until the last convolution layer. These results are shown in Figure 9.

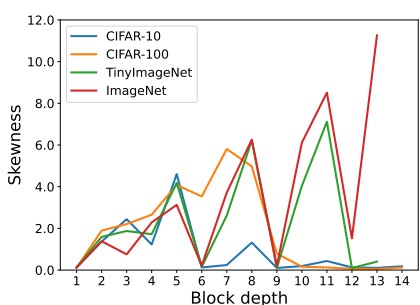

Figure 9: Asymmetric saturation of the Swap model on various dataset. There are no BN on fully connected layer in VGG16 for Tiny ImagaNet and ImageNet dataset, we only measure the skewness on a convolution layer.

## 7.2 RESULTS OF OTHER BOUNDED ACTIVATION FUNCTIONS

Our main investigations are based on the Tanh activation function. In this section, we test whether similar behavior is observed with other activation functions, such as LeCun Tanh (LeCun et al., 2012) and Softsign (Turian et al., 2009). In detail, we use the formula of LeCun tanh as follows $1.7159 \times tanh(\frac{2 \times input}{3})$. They are bounded and antisymmetric, just like Tanh. Softsign was proposed to prevent vanishing gradients by alleviating the saturation of neurons. It grows polynomially rather than exponentially, approaching its asymptotes much slower (Glorot & Bengio, 2010). LeCun Tanh has a gentle slope and a wider output range than Tanh. The asymmetric saturation caused by the Swap order occurs not only in Tanh but also in other activation functions. The shapes of these functions and layer skewness were shown in Appendix A.5. The Swap with Softsign and LeCun Tanh have improved performance compared to the Convention. It can be found in Table 2. When swapping, asymmetric saturation happens the least in Softsign, which makes it challenging to create a saturation state. Furthermore, the Softsign model shows lower performance than the Tanh model, which could generate more saturation with the most significant slope in the Swap, even though the Convention model had the highest performance.

## 8 CONCLUSION

In this work, we report that the Swap models perform better than the Convention models in many cases and analyze what brings about performance improvement. Asymmetric saturation at the channel level and sparsity induced by BN are two key factors explaining the better performance of the Swap models. With asymmetric saturation and normalization by BN, the final distributions generated by BN layers of the Swap models much resemble those by ReLU. This explains why the Swap models outperform the Convention models and often show results comparable to the ReLU models.

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

Table 3: Training hyperparameters of the VGG16 Tanh models

| | Convention | | | | Swap | | | |
|---|---|---|---|---|---|---|---|---|
| | CIFAR-10 | CIFAR-100 | Tiny ImageNet | ImageNet | CIFAR-10 | CIFAR-100 | Tiny ImageNet | ImageNet |
| Training Epochs | 200 | 200 | 200 | 100 | 200 | 200 | 200 | 100 |
| Learning Rate | 0.1 | 0.01 | 0.01 | 0.01 | 0.01 | 0.01 | 0.01 | 0.01 |
| Learning Rate Drop | 100, 150 | 100, 150 | 100, 150 | 30, 60 | 100, 150 | 100, 150 | 100, 150 | 60, 90 |
| Weight Decay | 0.0001 | 0.0005 | 0.001 | 0.0001 | 0.001 | 0.0005 | 0.001 | 0.001 |
| Batch Size | 128 | 128 | 128 | 256 | 128 | 128 | 128 | 256 |

Table 4: Training hyperparameters of the VGG16 ReLU models

| | Convention | | | | Swap | | | |
|---|---|---|---|---|---|---|---|---|
| | CIFAR-10 | CIFAR-100 | Tiny ImageNet | ImageNet | CIFAR-10 | CIFAR-100 | Tiny ImageNet | ImageNet |
| Training Epochs | 200 | 200 | 200 | 100 | 200 | 200 | 200 | 100 |
| Learning Rate | 0.01 | 0.01 | 0.1 | 0.1 | 0.01 | 0.01 | 0.01 | 0.01 |
| Learning Rate Drop | 100, 150 | 100, 150 | 100, 150 | 30, 60 | 100, 150 | 100, 150 | 100, 150 | 60, 90 |
| Weight Decay | 0.001 | 0.005 | 0.0001 | 0.0001 | 0.001 | 0.005 | 0.001 | 0.0005 |
| Batch Size | 128 | 128 | 128 | 256 | 128 | 128 | 128 | 256 |

Table 5: Training hyperparameters of the MobileNet Tanh models

| | Convention | | | | Swap | | | |
|---|---|---|---|---|---|---|---|---|
| | CIFAR-10 | CIFAR-100 | Tiny ImageNet | ImageNet | CIFAR-10 | CIFAR-100 | Tiny ImageNet | ImageNet |
| Training Epochs | 200 | 200 | 200 | 100 | 200 | 200 | 200 | 100 |
| Learning Rate | 0.1 | 0.1 | 0.01 | 0.1 | 0.1 | 0.1 | 0.1 | 0.1 |
| Learning Rate Drop | 100, 150 | 100, 150 | 100, 150 | 30, 60 | 100, 150 | 100, 150 | 100, 150 | 60, 90 |
| Weight Decay | 0.0001 | 0.0005 | 0.0001 | 0.0001 | 0.0001 | 0.0005 | 0.0001 | 0.0001 |
| Batch Size | 128 | 128 | 128 | 256 | 128 | 128 | 128 | 256 |

Table 6: Training hyperparameters of the MobileNet ReLU models

| | Convention | | | | Swap | | | |
|---|---|---|---|---|---|---|---|---|
| | CIFAR-10 | CIFAR-100 | Tiny ImageNet | ImageNet | CIFAR-10 | CIFAR-100 | Tiny ImageNet | ImageNet |
| Training Epochs | 200 | 200 | 200 | 100 | 200 | 200 | 200 | 100 |
| Learning Rate | 0.01 | 0.01 | 0.01 | 0.01 | 0.01 | 0.01 | 0.01 | 0.1 |
| Learning Rate Drop | 100, 150 | 100, 150 | 100, 150 | 30, 60 | 100, 150 | 100, 150 | 100, 150 | 60, 90 |
| Weight Decay | 0.001 | 0.005 | 0.005 | 0.0001 | 0.001 | 0.005 | 0.005 | 0.0001 |
| Batch Size | 128 | 128 | 128 | 256 | 128 | 128 | 128 | 256 |

# A    APPENDIX

## A.1    TRAINING HYPERPARAMETER

The hyperparameters used in training are shown in Table 3, 4, 5, 6. We sweep the learning rate and weight decay hyperparameter. The learning rate was 0.1 and 0.01. For CIFAR and Tiny-ImageNet datasets, we trained models with a batch size of 128, and the learning rate was reduced by one-tenth at 100 and 150 of the total 200 epochs, and we swept 4 weight decay of 0.005, 0.001, 0.0005, and 0.0001. For ImageNet datasets, we trained models with a batch size of 256, and the learning rate was reduced by one-tenth at 30 and 60 of the total 100 epochs, and we swept 3 weight decay of 0.001, 0.0005, and 0.0001. We chose the best averaged-accuracy model for the 3 random seeds and averaged the values of these three models for all measurements for analysis. Because of the computation issue, we only use 1 seed for ImageNet dataset with early stopping.

## A.2    NO BN

We also compare the saturation and skewness between the Convention model and the model without BN, we call this "NoBN" model. As shown in Figure 10, asymmetric saturation also occurs in the model without BN, we call this "NoBN" model. However, the NoBN model can not utilize the advantages of batch normalization (e.g., high learning rate), it shows low test accuracy than the Convention model even though asymmetric saturation exists compared to the Convention model. The accuracy of the Convention model is 64.84% and the accuracy of the NoBN model is 61.06%.

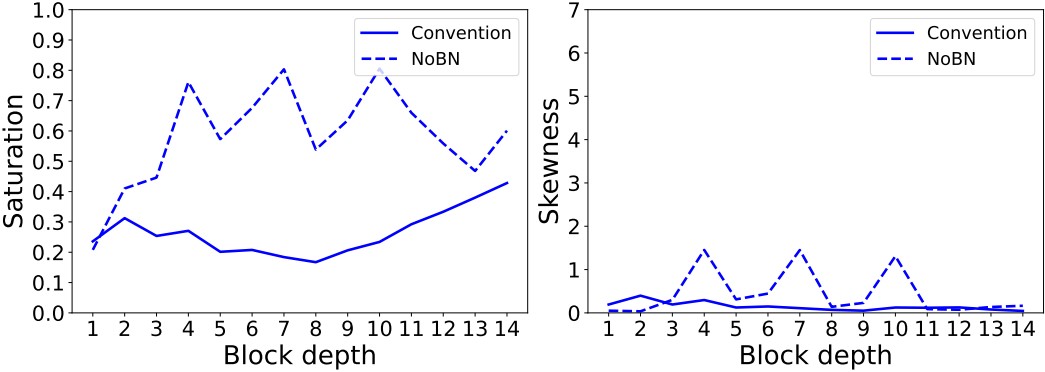

Figure 10: Layer saturation (left) and skewness (right) of the Convention VGG and the NoBN VGG model trained on CIFAR-100.

## A.3 SATURATION AND SKEWNESS MEASUREMENT VALUES

Our saturation metric becomes 0 when the distribution is gathered to 0, and it increases as the elements in the distribution close to the maximum expression range. For the uniform distribution, the degree of saturation was measured at 0.5. The measurement on different distributions can be found in Figure 11 (left). Skewness is the metric for measuring the asymmetry of the distribution. Skewness is calculated as 0 when the distribution is symmetric, and it increases as the asymmetry increase. We calculate the absolute on skewness in our asymmetry metric, thus the increases are regardless of the direction. The measurement on different distributions can be found in Figure 11 (right).

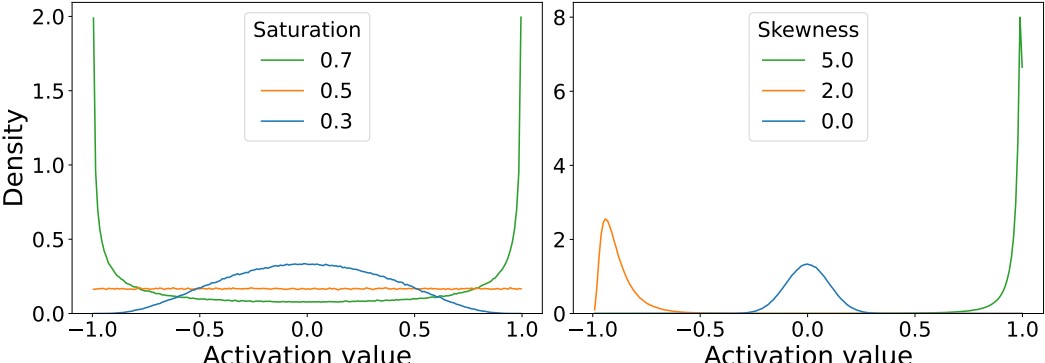

Figure 11: The degree of saturation on different distributions (left) and the degree of skewness on different distributions (right)

## A.4 THE EFFECTS OF THE MEAN AND STANDARD DEVIATION OF INPUT DISTRIBUTION ON TANH

The mean and variance of input distribution on Tanh affect the asymmetry of Tanh output. The skewness of Tanh output depends on the mean, and standard deviation can be found in Figure 12. The maximum skewness of varied mean distribution is increased on the increase of mean. However, the maximum skewness does not align with the input standard deviation increases. The skewness decreases not only the small input standard deviation but also the large input standard deviation. Additionally, in the same mean condition, a decrease in standard deviation from the maximum skewness point more rapidly decreases the skewness than an increase in standard devation.

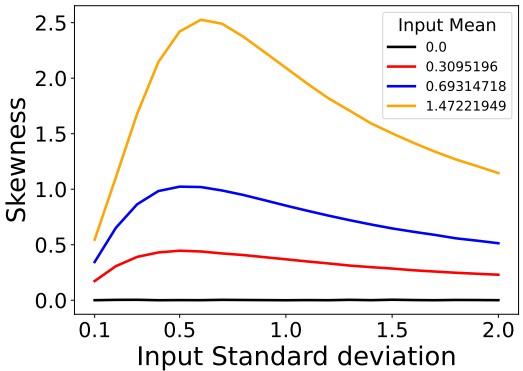

Figure 12: The skewness of Tanh output depend on the mean and standard deviation of Tanh input

### A.5 Skewness Tendencies on Various Activation Functions

The key to the success of the Tanh model with the Swap order is asymmetric saturation. We show that asymmetric saturation also appears in the other bounded activations, such as LeCun Tanh and SoftSign. The Conv model with the 3 types of activation functions shows low layer-wise skewness. The skewness is less than 1 over the overall layer. However, a significant skewness increment arises when the Swap order is applied. The SoftSign shows a minor improvement in skewness due to its property of preventing saturation.

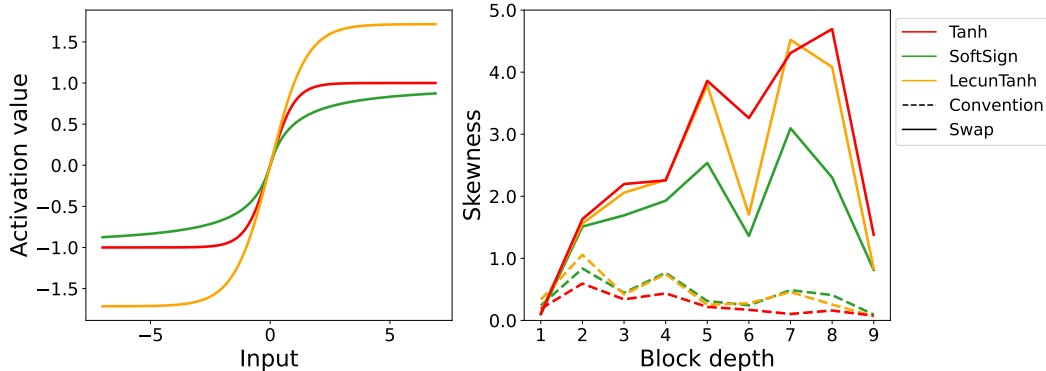

Figure 13: Shapes of activation functions (left) and skewness tendency of different activation functions (right), dashed line represents the Convention model and the solid line represents the Swap model.

### A.6 Best Depth Model Searching on VGG16

To find an appropriate model for CIFAR, we measured the accuracy of models without the last convolution layers of VGG16. We train them from scratch using VGG16's training hyperparameters. The accuracy gradually increases until the VGG16_11 model, and decreases after that. The results are shown in Table 7. One thing to note is that the omitted layers have a low skewness in the VGG16 model. The layer-wise skewness considerably decrease after the 8th block, which is the same number of convolution layers in the best performance model. The layer-wise skewness is shown in Figure 14.

Table 7: Performance of shortened Swap VGG16 models. The number of removed convolution layers in the VGG16_n model is the difference between 16 and n.

|          | VGG16 | VGG16_15 | VGG16_14 | VGG16_13 | VGG16_12 | VGG16_11 | VGG16_10 | VGG16_9 | VGG16_8 |
|----------|-------|----------|----------|----------|----------|----------|----------|---------|---------|
| Accuracy | 72.17 | 73.02    | 73.48    | 73.85    | 73.76    | 73.92    | 72.57    | 70.91   | 70.69   |

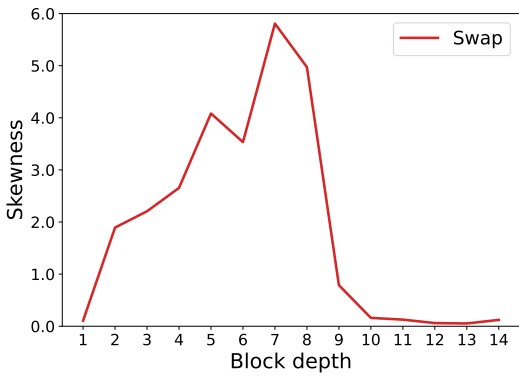

Figure 14: Skewness of the layers in the original VGG16 models

A.7   THE INPUTS OF WEIGHT LAYER AND THE GRADIENTS OF TANH AND WEIGHT

The vanishing gradients problem is inevitable when excessive saturation occurs. However, the Swap model can alleviate the gradient vanishing problem. The forward propagation among the convolution and Tanh layers in the Swap model is as follows: $\mathbf{y} = W\mathbf{x}$, $\mathbf{a} = Tanh(\mathbf{y})$. Here, $\mathbf{x}$ is a hwc-by-1 vector, and W is a d-by-n matrix, where $h$ is the height, $w$ is the width, $c$ is the number of channels, $d$ is the number of filters, and $n$ is the size of column $x$, i.e., $n = hwc$. In backpropagation, the gradient of W is obtained by the x of the corresponding dimension element. As a result, the larger x can solve the vanishing gradients problem. The Conv block's output is Tanh's output in the range of [-1, 1], while the Swap block's output can have a larger value since BN has no limit. A vanishing gradient occurs at Tanh of the Swap model in the experiment. However, it is alleviated on the gradient of convolution weight due to the large x, and shows a similar scale to the gradients of convolution weight in the Conv model. In the gradient on the shallow layers, the backpropagation gradients on Tanh of the Swap model are smaller than those of the Conv model. On the other hand, the Swap model has a larger scale of $\mathbf{x}$ than the Conv model. Thus, Conv and Swap models have a similar scale when looking at the gradient of the convolution weight.

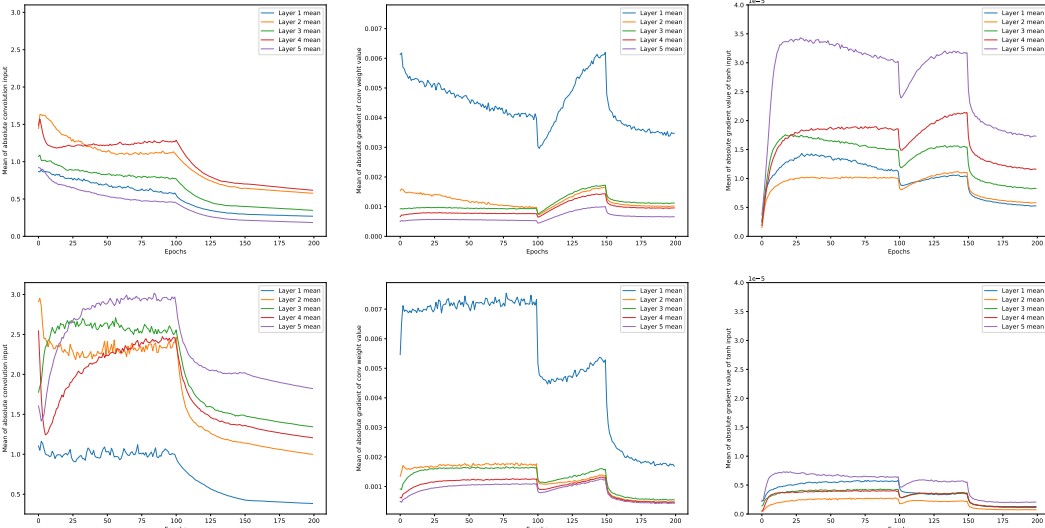

Figure 15: Plots for mean of absolute value of Convolution input(left) and mean of absolute gradient of Convolution weight value(center) and mean of absolute gradient of tanh input value(right) in the Convention(top) model and the Swap(bottom) model

## A.8    Learning Curve of Conv and Swap models

Both models were trained with the same hyperparameters. At the beginning of training, the training loss of the Swap model decreases faster than that of the Conv model, but when training is complete, the training losses of the two models become almost the same. However, through the validation loss, we can see that the Swap model has better generalization ability. The training loss is shown in the Figure 16 and the test loss is shown in the Figure 17.

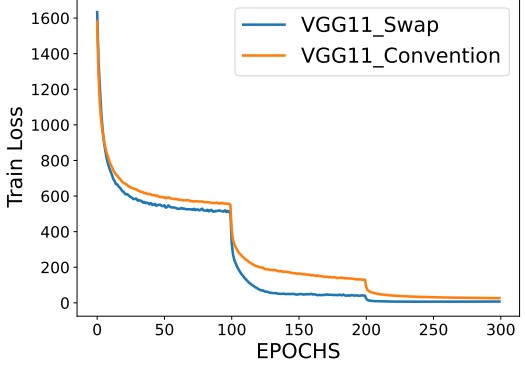

Figure 16: Training loss of Conv and Swap models.

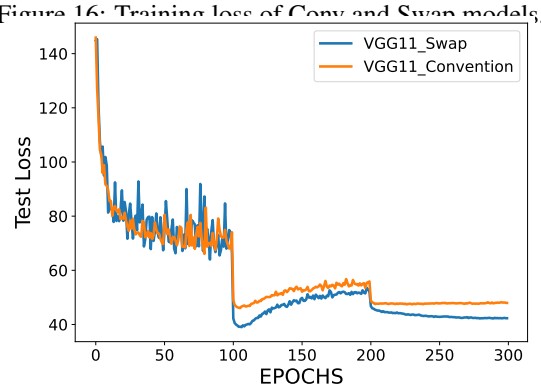

Figure 17: Test loss of Conv and Swap models.

## A.9    Relation between performance and sparsity for large affine parameters

We followed the As the size of the weight decay applied to the affine parameters increased, the sparsity decreased. Accordingly, it was confirmed that the performance also decreased.

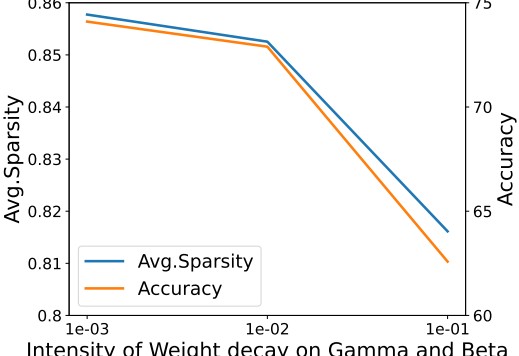

Figure 18: Accuracy drops as the sparsity decreases for large affine parameters.

## A.10    Proof of properties for sparsity metrics of inverse saturation

$G^l$ is a vector [g1, g2, g3, ..., gN] $\hat{G}^l$ is a vector [g1, g2, g3, ..., gN]

Theorem 1.1: $S$ satifies $S(\alpha G^l) = S(G^l), \forall \alpha \in \mathbb{R}, \alpha > 0$.

Proof: scaling the $G^l$ also scale the $\tilde{g}^l$.

$$\therefore \hat{G}^l(\alpha G^l) = \frac{\alpha |G^l|}{\alpha \widetilde{G^l}} = \frac{G^l}{\tilde{g}^l} = \hat{G}^l(G^l)$$

Theorem 1.2: $S$ satifies $S(\alpha + G^l) < S(G^l), \alpha \in \mathbb{R}, \alpha > 0$ (We also exclude the case mentioned in Hurley & Rickard (2009) that all elements of $G^l$ are the same.)

Proof:
$$S(G^l + \alpha) = \frac{\Sigma_{i=1}^N g_i^l + N\alpha}{N\tilde{g}^l + N\alpha}$$

if $N\tilde{g}^l > \Sigma_{i=1}^N g_i^l$ then $\frac{\Sigma_{i=1}^N g_i^l + N\alpha}{N\tilde{g}^l + N\alpha} > \frac{\Sigma_{i=1}^N g_i^l}{N\tilde{g}^l}$

$\therefore S(\alpha + G^l) < S(G^l)$

Theorem 1.3: $S$ satifies $S(G^l) = S(G^l||G^l||...||G^l)$

($||$ is concatenation)

Proof: We define $concat(X, t)$ which means concatenate vector X as t times. Then $S(concat(G^l, t)) = \frac{t\Sigma_{i=1}^N \hat{g}_i^l}{tN} = S(G^l)$

Theorem 1.4: $S$ satifies $\forall i \exists \beta = \beta_i > 0$, such that $\forall \alpha > 0$:
$$S([g_1^l...g_i^l + \beta + \alpha...]) > S([g_1^l...g_i^l + \beta...])$$
We choose sufficiently large $\beta$ that $|g_i^l| + \beta > \tilde{g}^l$. Let assume that $S([g_1^l...g_i^l + \beta + \alpha...]) \leq S([g_1^l...g_i^l + \beta...])$.
Then

$$1 - \frac{\Sigma_{k=1}^N g_k^l + \beta + \alpha}{N(g_i^l + \beta + \alpha)} \leq 1 - \frac{\Sigma_{k=1}^N g_k^l + \beta}{N(g_i^l + \beta)}$$
$$\frac{\Sigma_{k=1}^N g_k^l + \beta + \alpha}{N(g_i^l + \beta + \alpha)} \geq \frac{\Sigma_{k=1}^N g_k^l + \beta}{N(g_i^l + \beta)}$$
$$\frac{\Sigma_{k \neq i} g_k^l + g_i^l + \beta + \alpha}{g_i^l + \beta + \alpha} \geq \frac{\Sigma_{k \neq i} g_k^l + g_i^l + \beta}{g_i^l + \beta}$$
$$\frac{\Sigma_{k \neq i} g_k^l}{g_i^l + \beta + \alpha} \geq \frac{\Sigma_{k \neq i} g_k^l}{g_i^l + \beta}$$
$$\frac{1}{g_i^l + \beta + \alpha} \not\geq \frac{1}{g_i^l + \beta}$$
$$\therefore S([g_1^l...g_i^l + \beta + \alpha...]) > S([g_1^l...g_i^l + \beta...]).$$

Theorem 1.5: $S$ satifies $S(G^l||0) > S(G^l)$

Proof:
$$1 - \frac{\Sigma_{k=1}^N \hat{g}_k^l}{N + 1} > 1 - \frac{\Sigma_{k=1}^N \hat{g}_k^l}{N}$$

## A.11 ALGORITHMS

For more details, we take channel-wise summation with respect to batchs, but we divide the summation value by $D$ and accumulate as batch statistics. Because the whole step is same as taking average with respect to total sample, we can divide by the total size first and sum all values as batchs later. We follow this step due to the memory usage.

---

**Algorithm 1:** Calculating skewness over the layers

---

**Input:** $x_s(s = 1, 2, ..., S) = mini-batch \in \mathbb{R}^{b_s \times H \times W}$
      $N = total\,number\,of\,samples$
**Output:** $skewness \in \mathbb{R}^L$

1 **Function** `Statistics`($X$,$mean$,$variance$,$skewness$,$condition$)**:**
2    $X = \{X^1, X^2, ..., X^L\}$.
3    **for** $l = 1, 2, ..., L$ **do**
4       $C^l, b_s, h^l, w^l$ is the shape of $X^l$
5       $B = b_s \times h^l \times w^l, \quad D = N \times h^l \times w^l$
6       # $\mathbb{R}^{C^l \times B} \leftarrow \mathbb{R}^{C^l \times b_s \times h^l \times w^l}$
7       $block^l \leftarrow flat(X^l)$ by channels
8       **if** $condition\ is\ mean$ **then**
9          # $\mathbb{R}^{C^l} \leftarrow \mathbb{R}^{C^l \times B}$
10          1. $batch\_statistic^l \leftarrow$ sum of $block^l$ along the channels
11          2. $batch\_statistic^l \leftarrow \frac{batch\_statistic^l}{D}$
12          3. $mean^l$ += $batch\_statistic^l$
13       **end**
14       **if** $condition\ is\ variance$ **then**
15          # $\mathbb{R}^{C^l} \leftarrow \mathbb{R}^{C^l \times B}$
16          1. $batch\_statistic^l \leftarrow$ sum of $(block^l - mean^l)^2$ along the channels
17          2. $batch\_statistic^l \leftarrow \frac{batch\_statistic^l}{D}$
18          3. $variance^l$ += $batch\_statistic^l$
19       **end**
20       **if** $condition\ is\ skewness$ **then**
21          # $\mathbb{R}^{C^l} \leftarrow \mathbb{R}^{C^l \times B}$
22          1. $batch\_statistic^l \leftarrow$ sum of $\frac{(block^l - mean^l)^3}{\sqrt{variance^{l^3}}}$ along the channels
23          2. $batch\_statistic^l \leftarrow \frac{batch\_statistic^l}{D} \times \frac{\sqrt{D \times (D-1)}}{D-2}$
24          3. $skewness^l$ += $batch\_statistic^l$
25       **end**
26    **end**
27 **end**
28 $mean = \{mean^1, mean^2, ..., mean^L\}$.
29 $variance = \{variance^1, variance^2, ..., variance^L\}$.
30 $skewness = \{skewness^1, skewness^2, ..., skewness^L\}$.
31 **for** $s = 1, 2, ..., S$ **do**
32    $X \leftarrow getactivation(x_s)$ : get activation output values over the layers.
33    update $mean$ with `Statistics` ($X$,$mean$,$condition \leftarrow mean$)
34 **end**
35 **for** $s = 1, 2, ..., S$ **do**
36    $X \leftarrow getactivation(x_s)$ : get activation output values over the layers.
37    update $variance$ with `Statistics` ($X$,$mean$,$variance$,$condition \leftarrow variance$)
38 **end**
39 **for** $s = 1, 2, ..., S$ **do**
40    $X \leftarrow getactivation(x_s)$ : get activation output values over the layers.
41    update $skewness$ with
     `Statistics` ($X$,$mean$,$variance$,$skewness$,$condition \leftarrow skewness$)
42 **end**
43 # $\mathbb{R} \leftarrow \mathbb{R}^{C^l}$
44 $skewness^l \leftarrow$ Average of absolute of each channel values in $skewness^l$ along the layers.
45 **return** $skewness$

---

---

**Algorithm 2:** Calculating (Empirical) saturation over the layers

---

**Input:** $x_s(s = 1, 2, ..., S) = mini - batch \in \mathbb{R}^{b_s \times H \times W}$
$\quad\quad\quad N$ = total number of samples
$\quad\quad\quad saturation\_type$ = empirical or not
$\quad\quad\quad activation type is$ LeCun or not.
**Output:** $saturation \in \mathbb{R}^L$

1 **if** *saturation_type is empirical* **then**
2 $\quad$ $upper \leftarrow$ channel-wise maximum absolute value
3 **else if** *activation_type is $LeCun\_tanh$* **then**
4 $\quad$ $upper \leftarrow 1.7159$
5 **else**
6 $\quad$ $upper \leftarrow 1$
7 **end**

8 $saturation = \{saturation^1, saturation^2, ..., saturation^L\}.$
9 **for** $s = 1, 2, ..., S$ **do**
10 $\quad$ **if** *saturation_type is empirical* **then**
11 $\quad\quad$ $X \leftarrow getblock(x_s)$ : get block output values over the layers.
12 $\quad$ **end**
13 $\quad$ **else**
14 $\quad\quad$ $X \leftarrow getactivation(x_s)$ : get activation output values over the layers.
15 $\quad$ **end**
16 $\quad$ $X = \{X^1, X^2, ..., X^L\}.$
17 $\quad$ **for** $l = 1, 2, ..., L$ **do**
18 $\quad\quad$ $C^l, b_s, h^l, w^l$ is the shape of $X^l$
19 $\quad\quad$ $B = b_s \times h^l \times w^l, \quad D = N \times C^l \times h^l \times w^l$
20 $\quad\quad$ # $\mathbb{R}^{C^l \times B} \leftarrow \mathbb{R}^{C^l \times b_s \times h^l \times w^l}$
21 $\quad\quad$ 1. $block^l \leftarrow flat(X^l)$ by channels.
22 $\quad\quad$ 2. Take absolute to $block^l$.
23 $\quad\quad$ 3. $block^l \leftarrow \frac{block^l}{upper}$.
24 $\quad\quad$ # $\mathbb{R} \leftarrow \mathbb{R}^{C^l \times B}$
25 $\quad\quad$ 4. $sum \leftarrow$ sum of all values of $block^l$.
26 $\quad\quad$ 5. $saturation_s^l \leftarrow \frac{sum}{D}$.
27 $\quad\quad$ 6. $saturation^l += saturation_s^l$.
28 $\quad$ **end**
29 **end**
30 **return** $saturation$

---

