# OpenReview forum: "Batch Normalization and Bounded Activation Functions"
_ICLR.cc/2023/Conference — Submitted to ICLR 2023_

### Official Review · Reviewer_Qoao · 2022-10-22

**Confidence:** 4
**Correctness:** 2
**Technical Novelty And Significance:** 3
**Empirical Novelty And Significance:** 2
**Recommendation:** 5

**Clarity, Quality, Novelty And Reproducibility:**

The clarity is somewhat unclear and I have concerns on the main claims, the quality and novelty is somewhat good. I believe the experiments can be somewhat reproduced based on the descriptions of this paper.

**Strength And Weaknesses:**


**Strengths:**

1. normalization and nonlinear (activation) layers are basic layer/module in DNNs, and it is a good plus to investigate their interaction. This paper investigates How BN interact with bounded activation and shows several interesting observations，e,g., the activation values
of each channel of Swap model is asymmetrically saturated.

2. The view in investigating the correlation among saturation (especially the channel-wise saturation), sparsity and performance is new to me, and this paper also provides quantitative metric to evaluate the saturation and sparsity.



**Weaknesses:**

**1. One big concern for me is that some claims are not clearly clarified or rigorous.**

(1) In Introduction section, I donot understand this claim “In contrast, the one-sided property of asymmetric saturation causes at least half of the sample values after normalization to be almost zero, allowing the Swap model to have even higher sparsity than the Convention model”.  I find the only support is from Section 5.1. Indeed I donot understand why “The asymptotic values of combined Tanh with normalization operation are $\frac{+1-\hat{\mu}}{\hat{\sigma}}$ and $\frac{+1+\hat{\mu}}{\hat{\sigma}}$.”?  Can this paper provide further clarification? Besides, I understand “$\hat{\mu}$ becomes around -1 or 1 value”, but why $\hat{\sigma}$ is calculated as an appropriate size to produce a high skewness”? and what is a appropriate size?

(2) In Section 3.2, “This is counterintuitive as excessive saturation is considered an undesirable situation in the previous works” is somewhat misleading. I believe the experiments are based on the model at the end of training. I believe the “excessive saturation is considered an undesirable situation in the previous works” is for the initial training stages (the model has not learned information from the datasets), but it is not undesirable in the end of the training. Indeed, a good model with high confidence to the prediction may have excessive saturation neurons.

(3) The statement in Section 3 “When training a neural network with bounded activation functions with a center of the function at the origin, the output increases due to the weight gradually increasing.” is not rigorous. Does sigmoid (bounded activation) has a center of the function at the origin? It is also has likelihood that the bounded activation is saturated in the initial training, which depends how the model is initialized. Furthermore, why the weight gradually increasing? I think it also depends on the how much the weight decay is used and (for a model with weight decay)?

(4) It seems to be contradictory base on the statement “In short, higher saturation decreases sparsity” in Section 5.2 and the observation that Swap model has high saturation, but why the experimental results show that Swap model has higher sparsity? Is any wrong understanding of me?

(5) The observations seem to be not uniformly hold over all layers of VGG, from Figure 3 and 4. Why the activations of Swap model after the 10 th layer has lower saturation and almost the same Skewness, compare to the Conventional model, in Figure 3 and 4? That is not consistent to the observations of Swap model before the 10th layer.

**2. I also have concern on the experimental setup and analysis.**

(1) Considering this statement “Because Tanh has non-linearity in everyplace except the origin, it can not follow the design of residual connection proposed in He et al. (2016). Thus, we choose architectures where a skip connection does not exist.”, does this mean that the MobileNet used in this paper has no residual connection in its ResBlock? If yes, this paper only performs experiments on the VGG-like (feed forward neural network), but not for the ResNet-like network.  By the way, I donnot understand why residual connection is not allowed for Tanh? Does this design cannot well trained or the model cannot be conducted by design?

(2) In Figure 7, it is true that the increase in the models sparsity and accuracy are highly correlated. But I am wondering whether this results still hold by further increasing the weight decay on the affine parameters (the current maximum weight decay is $5e^{-4}$)? Intuitively, further increasing the weight decay on the affine parameters will further increase the sparsity, based on the statements shown in this paper. Does it further improve the accuracy of the network? What is the sparsity and accuracy, if we use a weight decay on the affine parameters of $1e^{-3}$, $1e^{-2}$, $1e^{-1}$?

(3) Even though Swap model using BN after bounded activation functions performances better than conventional model, one problem is that all the swap models using bounded activation functions shown in Table 1 has significantly lower accuracy compared to the conventional model using ReLU (the widely used unbounded activations). Based on this, I cannot well recognize the contributions of this paper, from the perspective of practice. It is better to provide the results that a swap model using BN+bounded activation has better performance than the conventional model using BN+ ReLU.


(4) It is good that this paper submit the code, but when I check how this paper calculate the saturation, skewness and sparsity in the code, I cannot find it. Do the authors tell me in which line and which file, the saturation, skewness and sparsity are calculated?


**Summary Of The Paper:**

This paper investigates how the order of batch normalization (BN) placed in the network affects the performance, when using the unbounded activation functions (e.g., Tanh). It shows that the Swap model (BN after the nonlinearity) using unbounded activation functions has significantly better performance than the conventual model (BN before the nonlinearity) using the same unbounded activation functions. It further shows activation values of each channel of Swap model is asymmetrically saturated, by looking into the output of channel-activations. It claims this asymmetrically saturated activations of Swap model increases the sparsity, which improves the performance of the model.

**Summary Of The Review:**

This paper provides a somewhat new view in investigating BN’s position with different nonlinearity. The observations are interesting but with unclear clarification. I personally have main concerns and currently tend to negative for this paper.

---

> ### Author Response · Authors · 2022-11-19
> **Response to Reviewer Qoao**
>
> We appreciate your valuable and constructive feedback.
>
> Weaknesses 1. One big concern for me is that some claims are not clearly clarified or rigorous.
>
> Weakness 1.1.  In Introduction section, I donot understand this claim “In contrast, the one-sided property of asymmetric saturation causes at least half of the sample values after normalization to be almost zero, allowing the Swap model to have even higher sparsity than the Convention model”. I find the only support is from Section 5.1. Indeed I donot understand why “The asymptotic values of combined Tanh with normalization operation are  +1−μ^σ^ and
> +1+μ^σ^.”? Can this paper provide further clarification? Besides, I understand “μ^ becomes around -1 or 1 value”, but why  σ^ is calculated as an appropriate size to produce a high skewness”? and what is a appropriate size?
>
> We clarify the description of Section 5.1 in the revised version. When asymmetric saturation occurs in the output of Tanh in the Swap model, most values are saturated to -1 or 1, indicating one-sided properties. The BN layer is placed right after Tanh in the Swap model. Thus, the normalization in BN shifts Tanh's outputs to a zero mean. [2] states that adding a constant to each coefficient decreases sparsity. They describe this intuitively:  "Give everyone a trillion dollars and the small differences in overall wealth are then negligible so everyone will have effectively the same wealth". Namely, the sparsity of vector x should be lower than vector x plus the constant positive a, except for the elements of x having the same wealth. Also, the inverse case, in which subtracting a constant from each coefficient increases sparsity, is also accepted. Making a zero mean on a one-sided distribution is subtracting a constant, specifically the estimated mean in the BN. Thus, normalization brings high sparsity.
>
> [2] Hurley, Niall, and Scott Rickard. "Comparing measures of sparsity." IEEE Transactions on Information Theory 55.10 (2009): 4723-4741.
>
> Weakness 1.2. In Section 3.2, “This is counterintuitive as excessive saturation is considered an undesirable situation in the previous works” is somewhat misleading. I believe the experiments are based on the model at the end of training. I believe the “excessive saturation is considered an undesirable situation in the previous works” is for the initial training stages (the model has not learned information from the datasets), but it is not undesirable in the end of the training. Indeed, a good model with high confidence to the prediction may have excessive saturation neurons.
>
> Based on some previous works [3, 4, 5], we considered excessive saturation to be poor in the learning process as well as in the initialization state. [3] states that when the model output is saturated, the network gives no indication of its confidence level. [4] regards the excessively saturated model as an imprecise model immune to further training. [5] shows that excessive saturation has a correlation with overfitting.
> We are willing to change our statement if there are references suggesting that excessive saturation is not undesirable in the end of the training.
>
> [3] LeCun, Yann A., et al. "Efficient backprop." Neural networks: Tricks of the trade. Springer, Berlin, Heidelberg, 2012. 9-48.
>
> [4] Rakitianskaia, Anna, and Andries Engelbrecht. "Measuring saturation in neural networks." 2015 IEEE symposium series on computational intelligence. IEEE, 2015.
>
> [5] Rakitianskaia, Anna, and Andries Engelbrecht. "Saturation in PSO neural network training: Good or evil?." 2015 IEEE Congress on Evolutionary Computation (CEC). IEEE, 2015.

---

> > ### Author Response · Authors · 2022-11-19
> > **Response to Reviewer Qoao (2)**
> >
> > Weakness 1.3. The statement in Section 3 “When training a neural network with bounded activation functions with a center of the function at the origin, the output increases due to the weight gradually increasing.” is not rigorous. Does sigmoid (bounded activation) has a center of the function at the origin? It is also has likelihood that the bounded activation is saturated in the initial training, which depends how the model is initialized. Furthermore, why the weight gradually increasing? I think it also depends on the how much the weight decay is used and (for a model with weight decay)?
> >
> > By “center of the function”, we meant the center of both domain and image of the function. In that sense, sigmoid does not have a center of the function at the origin and is not included in the activation functions of our focus.
> > In the descriptions in Section 3, our statement is based on the experiment in [6]. [6] shows that as the training proceeded, most of the activation values of Tanh got saturated sequentially, starting from the shallow layer. They also regard the phenomenon of saturation as occurring as the weight moves to a larger weight.
> >
> > [6] Glorot, Xavier, and Yoshua Bengio. "Understanding the difficulty of training deep feedforward neural networks." Proceedings of the thirteenth international conference on artificial intelligence and statistics. JMLR Workshop and Conference Proceedings, 2010.
> >
> > Weakness 1.4. It seems to be contradictory base on the statement “In short, higher saturation decreases sparsity” in Section 5.2 and the observation that Swap model has high saturation, but why the experimental results show that Swap model has higher sparsity? Is any wrong understanding of me?
> >
> > That is because we measure the saturation of Tanh output and the sparsity of block output as depicted in Figure 2. In the case of the Swap model, if the asymmetric saturation occurs in Tanh, then the BN layer pulls the mode of the activation distribution to near zero. Thus, much contradictorily, the higher asymmetric saturation brings about the higher sparsity in the Swap order model.
> >
> > Weakness 1.5.  The observations seem to be not uniformly hold over all layers of VGG, from Figure 3 and 4. Why the activations of Swap model after the 10 th layer has lower saturation and almost the same Skewness, compare to the Conventional model, in Figure 3 and 4? That is not consistent to the observations of Swap model before the 10th layer.
> >
> > As discussed in regard to Figure 9, it can be seen as a by-product of a large model trained on a small dataset. The asymmetric saturation tends to occur from the front layers and then progresses to the back layers only when necessary. Since the VGG model is proposed for the ImageNet dataset, it is overparameterized for CIFAR-100, which leads to lower asymmetric saturation in deeper layers.
> > To avoid confusion from this mismatch between the model and dataset, we revised the paper to use a fitting model for CIFAR. To determine the best model for CIFAR, we measured the accuracy as we removed the last convolution layers of VGG16 one by one from the end (the result is in the appendix). Interestingly, the more layers we remove, the better accuracy we get until we remove the last five layers. We suspect that these five layers are not much productive and just incur additional overhead. We call this model (VGG16 minus the last five convolution layers) VGG16_11 and use it for the analysis of the Swap model and the Convention model in the revised paper. In VGG16_11, every layer is fully utilized with high asymmetric saturation.

---

> > > ### Author Response · Authors · 2022-11-19
> > > **Response to Reviewer Qoao (3)**
> > >
> > >
> > > Weaknesses 2.I also have concern on the experimental setup and analysis.
> > > Weakness 2.1. Considering this statement “Because Tanh has non-linearity in everyplace except the origin, it can not follow the design of residual connection proposed in He et al. (2016). Thus, we choose architectures where a skip connection does not exist.”, does this mean that the MobileNet used in this paper has no residual connection in its ResBlock? If yes, this paper only performs experiments on the VGG-like (feed forward neural network), but not for the ResNet-like network. By the way, I donnot understand why residual connection is not allowed for Tanh? Does this design cannot well trained or the model cannot be conducted by design?
> > >
> > > Yes, The MobileNet model in our experiment is the first version of MobileNet. It doesn’t have a skip connection. The residual block was proposed by pointing out the degradation problem, which is generated by the difficulty of generating the identity mapping function as an optimal function due to multiple non-linear layers. Thus, they kept clean on shortcut connections or just performed linear projection when the dimensions were mismatched, i.e., y = F(x, {W_i})+x or y = F(x, {W_i})+W_sx. However, in the case of the Swap order, the activation function is placed in a shortcut connection when matching the dimensions, i.e., y=F(x, {W_i})+A(x). We thought this shortcut connection with non-linearity didn’t follow the design principle of the residual block, and even worse, Tanh can’t perform as an identity mapping function except for the origin. We, therefore, investigated the VGG-like networks first. Additionally, if we ignore this and apply Swap Tanh to the ResNet model, we show good performance. In our experiment, the accuracies of ResNet20 with the conventional order of ReLU and Tanh models trained on CIFAR-100 are 68.97% and 69.06%, respectively, and the ReLU and Tanh models with the Swap order accuracies are 69.06% and 69.17%, respectively. The Swap model with Tanh outperforms the Conv model with Tanh and shows comparable accuracy with the ReLU models.
> > >
> > > Weakness 2.2.  In Figure 7, it is true that the increase in the models sparsity and accuracy are highly correlated. But I am wondering whether this results still hold by further increasing the weight decay on the affine parameters (the current maximum weight decay is
> > > 5e−4)? Intuitively, further increasing the weight decay on the affine parameters will further increase the sparsity, based on the statements shown in this paper. Does it further improve the accuracy of the network? What is the sparsity and accuracy, if we use a weight decay on the affine parameters of  1e−3, 1e−2, 1e−1?
> > >
> > > Sparsity decreases as the intensity of weight decay increases (e.g., 1e-2, 1e-1). However, the accuracy of the model also decreases and shows high correlations with sparsity. The results are added in Appendix A.9.
> > >
> > > Weakness 2.4. It is good that this paper submit the code, but when I check how this paper calculate the saturation, skewness and sparsity in the code, I cannot find it. Do the authors tell me in which line and which file, the saturation, skewness and sparsity are calculated?
> > >
> > > We apologize for the inconvenience. It is in "util/logger.py". The skewness is calculated by the “calculate_skewness” method in the Logger class at line 61, and the saturation is calculated by the “calculate_saturation” method with an “empirical” argument set to False in the Logger class at line 185.  Lastly, the sparsity is calculated by leveraging the “calculate_saturation” method with an “empirical” argument set to True. It will return layer-wise saturation. We subtracted those values from 1 and averaged them to calculate the sparsity.

---

### Official Review · Reviewer_AJ9Z · 2022-10-24

**Confidence:** 2
**Correctness:** 2
**Technical Novelty And Significance:** 4
**Empirical Novelty And Significance:** 2
**Recommendation:** 5

**Clarity, Quality, Novelty And Reproducibility:**

Paper is well written. But, I think the approach is not very novel and the paper is thin on technical levels.

**Strength And Weaknesses:**

**Strength**
It is surprising that a simple swap of batch normalization and activation can enhance the performance of neural nets for bounded activations.

**Weakness**

- I am not convinced about the argument that asymmetric saturation enhances generalization. Are authors sure that the enhanced performance is not due to accelerated training? Is it possible to compare the convergence for two different neural architectures?
- For asymmetric saturation, is it possible to create saturation for conventional architectures; thereby enhancing their performance?
- How do stepsize, batch size, network width, depth, channel size affect the observation?
- Would be nice to analytically derive the reason for asymmetric saturation at initialization to make sure it is independent of the dataset.
- Is the swap only useful for image classification and convolution nets. Is the swap beneficial for MLPs as well?
- Does the swap for tanh activation outperform ReLU with conventional BN location?

**Summary Of The Paper:**

It is shown that putting batch normalization after activation function enhance optimization with SGD. Then authors experimentally compares the distribution of hidden representation for two different locations of batchnormalization. In particular, paper investigated the sparsity, and asymmetric saturation. A stark contrast is reported for asymmetric saturation when BN is placed before and after activation.

**Summary Of The Review:**

The paper reports an interesting observation in learning neural networks. However, this observation requires further justifications and investigations.

Thanks for the rebuttal response!

---

> ### Author Response · Authors · 2022-11-19
> **Response to Reviewer Aj9Z**
>
> We appreciate your valuable and constructive feedback.
>
> Weaknesses 1. I am not convinced about the argument that asymmetric saturation enhances generalization. Are authors sure that the enhanced performance is not due to accelerated training? Is it possible to compare the convergence for two different neural architectures?
>
> We added the training and test loss curves of Conv and Swap models in Appendix A.8. Although Swap is slightly faster in training loss convergence, the test loss of the Convention model does not decrease after epoch 200 even though its training loss keeps decreasing. We believe this confirms that the enhanced performance is not due to accelerated training.
>
> Weaknesses 2. For asymmetric saturation, is it possible to create saturation for conventional architectures; thereby enhancing their performance?
>
> Yes, there can be a little performance enhancement. However, it is less effective than asymmetric saturation. When we train the Conv Tanh model with applied weight decay on convolution layers and only the beta parameters in the BN layer, which can generate symmetric saturation rather than asymmetric saturation, it shows better performance than the Conv Tanh model; however, it shows lower performance than the NWDBN model, and even worse than the Conv model. The Conv, NWDBN, and NWDBN+beta-decay models have accuracies of 69.50%, 72.22%, and 70.42%, respectively.
>
> Weaknesses 6. Does the swap for tanh activation outperform ReLU with conventional BN location?
>
> We didn't intend to demonstrate that Tanh is better than ReLU. Our contribution is that we show Tanh's potential and propose a way to increase Tanh's utilization, which shows comparable performance with ReLU in some cases.

---

### Official Review · Reviewer_5V5C · 2022-10-25

**Confidence:** 3
**Correctness:** 3
**Technical Novelty And Significance:** 3
**Empirical Novelty And Significance:** 1
**Recommendation:** 5

**Clarity, Quality, Novelty And Reproducibility:**

Quality: borderline
Clarity: adequate

**Strength And Weaknesses:**

Strength:
- This paper investigates an interesting topic: the order of batch normalization(bn) and activation function(act), and gives a partial answer: placing bn after act gives better test performance when bounded activation functions were used.
- The logic chain is quite clear and easy to understand: this paper first reports the improvement of the Swap order, then analyzes the order from the point of saturation and asymmetric saturation, and found that the bn after asymmetric saturation can induce sparsity, which may be the reason for improvement.
- The finding of this paper is interesting. For a bounded act, the saturation is asymmetric, and it may assist the performance.

Weaknesses:
- The saturation metric seems not complete enough. Consider that if the maximum absolute value of G^l is small, which means that the values of G^l are centered around 0, the metric would be close to 1 also.
- Fig.3 and fig.4 only show high saturation and high skewness in shallow blocks, in deep blocks things seem changed. What does that mean? Is it means that in the deep block, we should use Convention order?
- Although the finding is interesting, the conclusion seems less practical. The performance of the Swap order with bounded acts cannot outperform the Convention order with the ReLU act. We still can use ReLU directly.
- Besides, (this may be a overclaim) since the authors report a counterintuitive example that the Swap order, which achieves better performance, accompanies saturation,  the author had better discuss it because intuitively the bounded act can induce gradient vanishment problems.

**Summary Of The Paper:**

This paper reports that when a bounded activation function is used, the Swap order of network block "conv/fc+act+bn" achieves performance better than the Convention order of block "conv/fc+bn+act". The authors find that the bounded activation function accompanies high saturation in each network block and asymmetry saturation in each channel within the block. The asymmetry saturation with batch normalization can induce high sparsity, which assists the generalization performance. Three metrics were proposed to measure saturation, asymmetry saturation (skewness), and sparsity respectively. Experiments show that when several bounded activation functions are used, Swap order performs better than Conventional order in the classification tasks, but still cannot outperform ReLU-based networks.

**Summary Of The Review:**

This paper discusses an interesting topic, but the analysis seems less comprehensive. I would like to rate it as borderline reject. If the author can clarify my problems, I am willing to consider changing my score.

---

> ### Author Response · Authors · 2022-11-19
> **Response to Reviewer 5V5C**
>
> We appreciate your valuable and constructive feedback.
>
> Weaknesses 1. The saturation metric seems not complete enough. Consider that if the maximum absolute value of G^l is small, which means that the values of G^l are centered around 0, the metric would be close to 1 also.
>
> Sparsity is scale-invariant, and relative values are important, not absolute values (Hurley & Rickard, 2009). Similarly, saturation should be scale invariant. If all elements have the maximum value, the saturation metric should be 1, whether the maximum value is small or large. That is, the situation you mentioned can occur and is very legitimate.
>
>
> Weaknesses 2. Fig.3 and fig.4 only show high saturation and high skewness in shallow blocks, in deep blocks things seem changed. What does that mean? Is it means that in the deep block, we should use Convention order?
>
> As discussed in regard to Figure 9, it can be seen as a by-product of a large model trained on a small dataset. The asymmetric saturation tends to occur from the front layers and then progresses to the back layers only when necessary. Since the VGG model is proposed for the ImageNet dataset, it is overparameterized for CIFAR-100, which leads to lower asymmetric saturation in deeper layers.
> To avoid confusion from this mismatch between the model and dataset, we revised the paper to use a fitting model for CIFAR. To determine the best model for CIFAR, we measured the accuracy as we removed the last convolution layers of VGG16 one by one from the end (the result is in the appendix). Interestingly, the more layers we remove, the better accuracy we get until we remove the last five layers. We suspect that these five layers are not much productive and just incur additional overhead. We call this model (VGG16 minus the last five convolution layers) VGG16_11 and use it for the analysis of the Swap model and the Convention model in the revised paper. In VGG16_11, every layer is fully utilized with high asymmetric saturation.
>
> Weaknesses 4. Besides, (this may be a overclaim) since the authors report a counterintuitive example that the Swap order, which achieves better performance, accompanies saturation, the author had better discuss it because intuitively the bounded act can induce gradient vanishment problems.
> We also agree that the vanishing gradients problem can be a concern when using Tanh if it has an excessive saturation. When we measure the absolute backpropagation gradients of Tanh on the Conv and Swap VGG16_11 models, the Swap model shows about 10 times smaller gradients than the Conv model. However, since the input of the convolution layer in the Swap model is the output of the BN, the absolute gradients of convolution weight are on similar scales to the Conv model, which is the input of the convolution layer and the output of Tanh. More specifically, the forward propagation among the convolution and Tanh layers in the Swap model is as follows: y=Wx, a=Tanh(y). Here, x is a hwc-by-1 vector, and W is a d-by-n matrix, where h is the height, w is the width, c is the number of channels, d is the number of filters, and n is the size of column x, i.e., n = hwc. In backpropagation, the gradient of W is obtained by the x of the corresponding dimension element. As a result, the larger x can solve the problem of vanishing gradients. The Conv block's output is Tanh's output in the range of [-1, 1], while the Swap block's output can have a larger value due to BN having no bound.  In the experiment, a vanishing gradient occurs in Tanh of the Swap model, but it is alleviated on the gradient on convolution weight due to the large x, and shows a similar scale to the gradients of convolution weight in the Conv model. The experiment is added in Appendix A.7.

---

### Official Review · Reviewer_DHjF · 2022-10-25

**Confidence:** 3
**Correctness:** 3
**Technical Novelty And Significance:** 2
**Empirical Novelty And Significance:** 3
**Recommendation:** 3

**Clarity, Quality, Novelty And Reproducibility:**

Even though the observation that the Swap model with a bounded activation is interesting, the arguments that connect the observation and other experiments to the conclusions are not convincing. It seems that the training details are well-provided enough to enable reproducibility.

**Strength And Weaknesses:**

### Strengths
- The authors discover that in the Swap model with a bounded activation, each feature map is saturated on one side of the asymptotic value of the bounded activation.


### Weaknesses
- It is a bit confusing whether the asymmetric saturation has a strong association with generalization performance.
    - Although there is another noticeable observation that the saturation is very low in higher block depths, this is not discussed at all.
    - Can you explain more how to exclude the possibility that low saturation at higher blocks or the combination of both could be a reason for better generalization?
- It is also confusing whether the sparsity has a strong association with generalization performance
    - Since the sparsity metric is $s^l = 1 - t^l$ where $t_l$ is the saturation metric, layerwise sparsity can be obtained from Figure 3. The relation between the sparsity of the Swap model and the Convention model is different depending on which layer is considered. In such case, it seems a bit of a stretch to draw a conclusion that the higher the sparsity is the better the generalization is.
    - In a sense, this contradicts with the authors' argument 'Our saturation metric can dismiss the channel properties due to the summarization of channels in the layer.'
    - Can we say that different sparsity distributions over layers with the same average sparsity will have similar generalization performance?
- The coverage of the analysis is a bit limited.
    - The analysis is claimed to be valid with bounded nonlinearity and without residual connection, excluding many widely used architectures. Also, it seems difficult to generalize or apply the claim of the paper to commonly used cases.
    - Even though it is subjective, it does not seem that the Swap model with Tanh performs comparably to the Convention model with ReLU.

### Questions
- Can you elaborate on ' Because Tanh has non-linearity in everyplace except the origin, it can not follow the design of residual connection proposed'?
    - Does that mean that Tanh has gradient 1 at the origin? What does it mean by nonlinear in everyplace?
    - What does it mean by 'following the design of residual connection'?
- In Table 1, with ReLu, the Convention model is better than the Swap model. Have you considered or performed a similar analysis to understand this reversed behavior?
- What is 'the center of the function'? This term is not defined precisely. The center of the domain of the function or the center of the image of the function?
- NWDBN is not explained until Figure 8 and is frequently used before Figure 8. Even though I guess that NWDBN may stand for No Weight Decay Batch Normalization, acronyms should be explained when it is first used.
- What is the formula of LeCun Tanh? There are many typos for LeCun Tanh.

**Summary Of The Paper:**

This paper conducts an empirical analysis of the interaction between batch normalization and bounded activation functions. Specifically, the paper compares the architecture using batch normalization after a bounded activation(Swap model) and the architecture using a bounded activation after batch normalization(Convention model). Motivated by the observation that the swap model outperforms the convention model significantly when a bounded activation is used, the authors designed experiments to identify the reasons for these performance differences. The paper shows that in terms of asymmetric saturation, the Swap model and the Convention model behave differently and argues that high sparsity induced from the asymmetric saturation has a strong association with the generalization performance.

**Summary Of The Review:**

It is interesting to know that with a bounded activation, the order between BN and the activation causes drastically different qualitative behavior. However, the presentation of the idea can be improved further by replacing vaguely defined terms and expressions. The arguments supporting the conclusions seem weak.

---

> ### Author Response · Authors · 2022-11-19
> **Response to Reviewer DHjF**
>
> We appreciate your valuable and constructive feedback.
>
> Weakness 1. It is a bit confusing whether the asymmetric saturation has a strong association with generalization performance.
> Weakness 1.1. Although there is another noticeable observation that the saturation is very low in higher block depths, this is not discussed at all.
> As discussed in regard to Figure 9, it can be seen as a by-product of a large model trained on a small dataset. The asymmetric saturation tends to occur from the front layers and then progresses to the back layers only when necessary. Since the VGG model is proposed for the ImageNet dataset, it is overparameterized for CIFAR-100, which leads to lower asymmetric saturation in deeper layers.
> To avoid confusion from this mismatch between the model and dataset, we revised the paper to use a fitting model for CIFAR. To determine the best model for CIFAR, we measured the accuracy as we removed the last convolution layers of VGG16 one by one from the end (the result is in the appendix). Interestingly, the more layers we remove, the better accuracy we get until we remove the last five layers. We suspect that these five layers are not much productive and just incur additional overhead. We call this model (VGG16 minus the last five convolution layers) VGG16_11 and use it for the analysis of the Swap model and the Convention model in the revised paper. In VGG16_11, every layer is fully utilized with high asymmetric saturation.
>
> Weakness 1.2. Can you explain more how to exclude the possibility that low saturation at higher blocks or the combination of both could be a reason for better generalization?
>
> First, the accuracy of VGG16 increases until the model becomes VGG16_11 in the experiment above. Second, there is no phenomenon of low saturation and skewness in higher-depth layers in VGG16_11.
>
> Weakness 2. It is also confusing whether the sparsity has a strong association with generalization performance
> Weakness 2.1. Since the sparsity metric is sl=1−tl where tl is the saturation metric, layerwise sparsity can be obtained from Figure 3. The relation between the sparsity of the Swap model and the Convention model is different depending on which layer is considered. In such case, it seems a bit of a stretch to draw a conclusion that the higher the sparsity is the better the generalization is.
>
> There was an unclear part in our paper. We set the maximum absolute value as a theoretical absolute value of the bounded activation function (e.g., for Tanh, we used 1 for G^l) and measured it per layer for measuring the saturation of the bounded activation function. On the other hand, we empirically measured the maximum absolute value per channel for the sparsity measure. In other words, we measured the channel-wise saturation for measuring the sparsity and averaged it. We made it clear in the revised version. In the previous paper [1] that showed the advantages of sparsity, the sparsity in the model existed before the layer where weighted summation was performed. Likewise, we tried to measure the sparsity in front of the weight layer.
>
> [1] Glorot, Xavier, Antoine Bordes, and Yoshua Bengio. "Deep sparse rectifier neural networks." Proceedings of the fourteenth international conference on artificial intelligence and statistics. JMLR Workshop and Conference Proceedings, 2011.
>
> Weakness 2.2. In a sense, this contradicts with the authors' argument 'Our saturation metric can dismiss the channel properties due to the summarization of channels in the layer.'
>
> When we vary the intensity of weight decay on affine parameters, there is a similar tendency on a similar averaged sparsity model. Also, the models with similar averaged sparsity show similar accuracy.
>
> Weakness 2.3. Can we say that different sparsity distributions over layers with the same average sparsity will have similar generalization performance?
>
> We did not propose a universal metric that can be used in all general situations (e.g., comparisons between different architectures and comparisons among models of different capacities). It was introduced to analyze the performance difference between Conv and Swap models. In our experimental environment, there were no cases in which the tendency of layer-wise sparsity was extremely different. Therefore, we regarded this metric as practicable to use for our analysis.

---

> > ### Author Response · Authors · 2022-11-19
> > **Response to Reviewer DHjF (2)**
> >
> >
> > Questions 1.1. Can you elaborate on ' Because Tanh has non-linearity in everyplace except the origin, it can not follow the design of residual connection proposed'?
> > Questions 1.2. Does that mean that Tanh has gradient 1 at the origin? What does it mean by nonlinear in everyplace?
> >
> > This means that Tanh can work as an identity mapping only near the origin (in the context of the discussion below).
> >
> > Questions 2. What does it mean by 'following the design of residual connection'?
> >
> > The residual block was proposed by pointing out the degradation problem, which includes the difficulty of generating the identity mapping function as an optimal function due to multiple non-linear layers. Thus, they kept clean on shortcut connections or just performed linear projection when the dimensions were mismatched, i.e., y = F(x, {W_i})+x or y=F(x, {W_i})+W_sx. However, in the case of the Swap order, the activation function is placed in a shortcut connection when matching the dimensions, i.e., y=F(x, {W_i})+A(x). We thought this shortcut connection with non-linearity didn’t follow the design principle of the residual block, and even worse, Tanh can’t perform as an identity mapping function except for the origin.
> >
> > Questions 3. In Table 1, with ReLu, the Convention model is better than the Swap model. Have you considered or performed a similar analysis to understand this reversed behavior?
> >
> > We have not analyzed the behavior deeply. It might be the Convention model is slightly better at handling the covariate shift problem than the Swap model. With bounded activation functions, the benefit from the asymmetric saturation seems to surpass this disadvantage.
> >
> > Questions 4. What is 'the center of the function'? This term is not defined precisely. The center of the domain of the function or the center of the image of the function?
> >
> > As you mentioned, the center of the function can be defined as both domain and image. When the center of the domain of the function and the center of the image of the function are both zero, it means that the center of the function is at the origin.
> >
> > Question 5. NWDBN is not explained until Figure 8 and is frequently used before Figure 8. Even though I guess that NWDBN may stand for No Weight Decay Batch Normalization, acronyms should be explained when it is first used.
> >
> > What you mentioned is correct. We corrected it in the revised version.
> >
> > Question 6. What is the formula of LeCun Tanh? There are many typos for LeCun Tanh
> >
> > We use 'lecun_tanh(x) = 1.7159 * tanh((2*x)/3)' as a LeCun Tanh activation function. Also, we corrected it in the revised version.

---

> > ### Comment · Reviewer_DHjF · 2022-12-14
> > **Response to the rebuttal**
> >
> > Thanks for the answers!
> >
> > - Can you also add figures of the skewness of VGG16_11? Figure 14 is on the skewness of the original VGG16. In order to argue that the (asymmetric) skewness is helpful for generalization, it will be more convincing if we observe high skewness in every layer of VGG16_11, as mentioned in the author's rebuttal. In the current revision, the skewness is from the original VGG16. If the figures for the skewness of VGG16_xx are provided, that would more strongly support the argument. I hope that the authors kept the model they trained, then providing these additional figures won't be too demanding. Moreover, in Figure 14, the skewness at the 9th and 10th layers is low but accuracy is the highest at VGG16_11. If more quantitative information (figures or numbers) is provided, this will be more strong support for the argument.
> > - Reading the authors' to another reviewer, I see "In that sense, sigmoid does not have a center of the function at the origin and is not included in the activation functions of our focus.". Even though sigmoid, as it is, does not have the center according to the authors' definition, but $sigmoid(x) -0.5$ has the center according to the authors' definition. It seems that this additional translation can be observed in the parameters of batch normalization. In this sense, it seems that this translated sigmoid, in turn, original sigmoid should be analyzable. Is there any other reason that sigmoid cannot be considered in the analysis?

---

### Decision · Program_Chairs · 2023-01-20

**Decision:**

Reject

**Justification For Why Not Higher Score:**

I am not fully convinced by the claim that there is a relation between the saturation and generalization. Reviewers also mentioned that the bounded activation function like tanh with Swap order can not outperform the unbounded activation function with convention order.

**Justification For Why Not Lower Score:**

N/A

**Metareview: Summary, Strengths And Weaknesses:**

This paper points out an intersting fact that Swap order, i.e., putting batch normalization (BN) after bounded activation function (BAF) can achieve better performance than putting BN before BAF  in image classification. However, I am not fully convinced by the claim that there is a relation between the saturation and generalization. Reviewers also mentioned that the bounded activation function like tanh with Swap order can not outperform the unbounded activation function with convention order. The rebuttal did not address these concerns. For these reasons, I recommend reject.